# Robust Visual-Aided Autonomous Takeoff, Tracking, and Landing of a Small UAV on a Moving Landing Platform for Life-Long Operation

**Pablo R. Palafox** [1,*] , **Mario Garzón** [2] , **João Valente** [3] **and Juan Jesús Roldán** [4] **and Antonio Barrientos** [4]

1   Computer Vision & Artificial Intelligence Group, Technical University Munich, Boltzmannstrasse 3, 85748 Garching, Germany
2   Univ. Grenoble Alpes, INRIA, Grenoble INP, 38000 Grenoble, France
3   Information Technology Group, Wageningen University & Research, 6708 PB Wageningen, The Netherlands
4   Centro de Automática y Robótica (UPM-CSIC), Universidad Politécnica de Madrid, José Gutiérrez Abascal, 2, 28006 Madrid, Spain
*   Correspondence: pablo.rodriguez-palafox@tum.de



**Featured Application: Autonomous takeoff, tracking, and landing maneuvers on a moving target with application to a fleet of robots with aerial and ground vehicles that need to operate for extended periods of time, as in Search and Rescue tasks.**

**Abstract:** Robot cooperation is key in Search and Rescue (SaR) tasks. Frequently, these tasks take place in complex scenarios affected by different types of disasters, so an aerial viewpoint is useful for autonomous navigation or human tele-operation. In such cases, an Unmanned Aerial Vehicle (UAV) in cooperation with an Unmanned Ground Vehicle (UGV) can provide valuable insight into the area. To carry out its work successfully, such as multi-robot system requires the autonomous takeoff, tracking, and landing of the UAV on the moving UGV. Furthermore, it needs to be robust and capable of life-long operation. In this paper, we present an autonomous system that enables a UAV to take off autonomously from a moving landing platform, locate it using visual cues, follow it, and robustly land on it. The system relies on a finite state machine, which together with a novel re-localization module allows the system to operate robustly for extended periods of time and to recover from potential failed landing maneuvers. Two approaches for tracking and landing are developed, implemented, and tested. The first variant is based on a novel height-adaptive PID controller that uses the current position of the landing platform as the target. The second one combines this height-adaptive PID controller with a Kalman filter in order to predict the future positions of the platform and provide them as input to the PID controller. This facilitates tracking and, mainly, landing. Both the system as a whole and the re-localization module in particular have been tested extensively in a simulated environment (Gazebo). We also present a qualitative evaluation of the system on the real robotic platforms, demonstrating that our system can also be deployed on real robotic platforms. For the benefit of the community, we make our software open source.

**Keywords:** robust autonomous landing; unmanned aerial vehicle; unmanned ground vehicle; multi-robot systems; Kalman filter; PID controller; re-localization module

---

## 1. Introduction

Robotics is increasingly taking on greater importance in our lives. One of the main areas where this can be perceived is Search and Rescue (SaR) tasks [1]. Robots designed for this kind of task,

known as SaR robots, must operate on many occasions in unknown environments, move over unstable surfaces, and face multiple difficulties in order to carry out their mission, e.g., obtaining a map of the environment to facilitate the subsequent intervention of the rescue brigades [2]. Using a single robot under such conditions poses big difficulties: whether it moves on the surface or flies nearby areas, there are intrinsic difficulties for each type of robot. Thus, by building heterogeneous teams of robotic platforms that can jointly operate in such scenarios, it is possible to bring about great benefits, since the shortcomings of each robot can be compensated with the strengths of the other [3,4].

Indeed, while aerial robots have the unique ability to obtain top views from the terrain and move without being hampered by the elements that may be found on the ground after a collapse, their reduced flight autonomy limits their operating time to a few tens of minutes. Moreover, their load capacity is generally less than 1 kg, which limits the type of sensors or equipment that can be deployed.

On the other hand, terrestrial robots are able to overcome, in general, the requirements of energy autonomy and payload. In addition, they can act as relays for communication systems, as well as provide high computing capabilities and data storage to the system. They have, however, limited mobility, especially in cluttered environments, such as narrow bridges or inclined planes. Additionally, their ability to obtain information about their environment may also be limited by their low height above the ground level and by the very elements of the scenario.

The literature contains multiple examples of successful collaboration of ground and aerial robots to carry out different missions: exploration in wide areas with obstacles [3]; precision farming for ground moisture sampling [5]; surveillance in complex environments using route optimization strategies [6]; and supporting aerial surveys in maritime environments [7], where the maritime robot acts as a mobile landing platform of the Unmanned Aerial Vehicle (UAV) when it has to perform an emergency landing, charge its batteries, or be picked up by an operator. All these examples prove the efficiency and benefit of building mixed robotic systems comprised of a terrestrial and an aerial robot for many different and complex tasks.

This work proposes a step towards obtaining such a joint team by developing a system that enables a UAV to: (1) take off autonomously from a landing platform attached to a Unmanned Ground Vehicle (UGV); (2) detect, localize, and follow the ground robot while in the air; and (3) land autonomously on the moving platform when required.

The proposed system differs from previous works by presenting a novel height-adaptive controller for tracking and landing. In essence, the behavior of a Proportional-Integral-Derivative (PID) controller is modified according to the UAV's distance to the landing platform along the vertical axis. By doing so, the performance and robustness of the system as a whole are largely increased.

Two different approaches to track and land robustly the UAV on the moving landing platform are presented: in the first variant, the aforementioned height-adaptive controller uses the current position of the landing platform as the target; the second approach extends the height-adaptive controller with a prediction algorithm (based on a Kalman filter) that predicts the future position of the platform and feeds it to the controller. This facilitates tracking and, more importantly, the autonomous landing of the aerial robot.

Another key novelty introduced in this work is the addition of a recovery and re-localization module for both tracking and landing. This further helps to increase the robustness of the system, because the UAV can re-detect the landing platform autonomously in case the latter disappears from the field of view of the UAV's camera or if the relative error between the landing platform and the UAV in the immediate moments before landing is greater than a threshold.

Furthermore, a novel finite state machine is presented in this work, which together with our re-localization module allows for life-long operation, as we will demonstrate in Section 4.

The proposed system in its two versions has been extensively tested on a realistic three-Dimensional (3D) simulated environment (Gazebo [8]) and deployed for qualitative evaluation on real robotic platforms. Figure 1 shows the proposed aerial-ground robot fleet. We employ Robotnik's Summit XL as the UGV and the Parrot's AR Drone 2.0 as the UAV. In the simulated environment, we

use the corresponding Robot Operating System (ROS) [9] packages, namely the *summit_xl_sim* and *tum_simulator* packages.

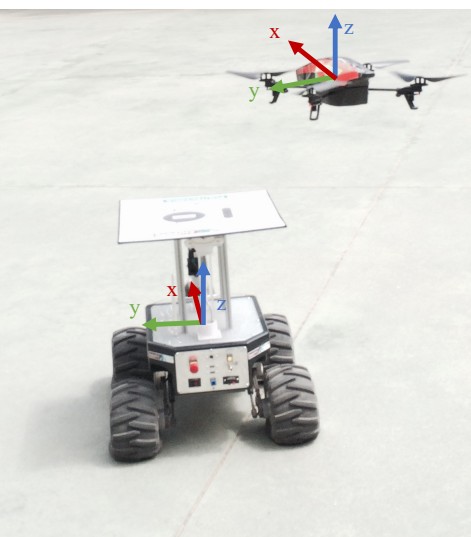

**Figure 1.** Cooperation between Unmanned Aerial Vehicles (UAVs) and Unmanned Ground Vehicles (UGVs) can greatly benefit Search and Rescue (SaR) tasks where both long-term operation and a wide aerial view are required. The UAV can travel on top of the UGV (possibly recharging its battery) and, when needed, take off, inspect the area, and land again autonomously.

This paper is organized as follows. Section 2 analyzes previous work. In Section 3, an overview of our system is presented, and the robotic frameworks and platforms used are described. Section 4 presents an extensive evaluation of the system in the simulated environment and qualitative tests in the real robotic platforms. Section 5 concludes this work.

## 2. Related Work

The development of drone-related applications has exploded in recent years. In particular, there is enormous interest in using these robots for detecting and monitoring terrestrial mobile objects using their on-board cameras. However, as previously mentioned, their low autonomy renders them unable to perform tasks of long duration. That is why much of the research so far has focused on landing UAVs on mobile platforms, giving them greater versatility, since in many scenarios, it is impossible to ensure a stationary landing area. A good overview of the research done in the development of vision-based autonomous landing systems, as well as the challenges in this field, can be found in [10].

Ling et al. [11] tried to solve the problem that arises when taking pictures of icebergs using drones launched from a ship. Traditionally, the aerial vehicle had to be rescued semi-manually between two operators: one would pilot the drone until it was close enough to the boat for a second operator to manually recover it, with the danger that this action entailed. Ling proposed a precision landing algorithm to eliminate completely the human participation in this type of situation, which uses a downward-facing camera to track a target on the landing platform and generates high quality relative pose estimates.

Lee et al. [12] focused on the use of vertical cameras and Image-Based Visual Servoing (IBVS) algorithms to track a platform in a two-dimensional space and perform a Vertical Take-Off and Landing (VTOL). They obtained the speed at which the platform moved, and then they used this information as a reference to perform an adaptive control of sliding movement. Compared to other vision-based control algorithms that reconstruct a complete 3D representation of the objective (which requires accurate depth estimates), the IBVS algorithms are computationally less expensive.

Prior to these two works, Saripalli and Sukhatme [13] worked with vision algorithms for the autonomous landing of a helicopter on a mobile platform. They used Hu's moments of inertia [14] for an accurate detection of the objective and a Kalman filter for tracking. Based on the output of the tracking algorithm, it was possible for them to implement a trajectory controller that ensured the landing on the mobile target.

The literature also contains some proposals with Model Predictive Control (MPC). Maces et al. [15] considered a mission with three phases (target detection, target tracking, and autonomous landing) that were modeled in a state machine. During the last two phases, an MPC is used for position control, whereas a PID controller is employed for altitude control. The system we present extends the state machine proposed by Maces with a key additional phase, namely a *recovery mode*. This new state increases the system's robustness by allowing the UAV to re-locate the landing platform autonomously in case the latter accidentally leaves the field of view of the drone's camera. Feng et al. [16] combined a vision-based target position measurement, a Kalman filter for target localization, an MPC for the guidance of the UAV, and an integral control for robustness. They tested their algorithms on a DJI M100 quadcopter and reached a maximum error of 37 cm with a platform moving at up to $12 \, \mathrm{m \, s^{-1}}$.

However, there are works that considered other techniques. Almeshal et al. [17] proposed a neural network to estimate the target position, as well as a PID controller to track it and perform landing, and validated it with a Parrot AR.Drone quadcopter. Finally, Yang et al. [18] developed a complete UAV autonomous landing system using a hybrid camera array (fish-eye and stereo cameras) and a state estimation algorithm based on motion compensation and tested them with multiple platforms (Parrot Bebop and DJI M100).

A common assumption in many of these systems is that the speed of the mobile target is low enough for the UAV to be able to land on it without compromising the integrity of both robotic platforms. However, experiments carried out by the German Aerospace Center (DLR) have demonstrated that it is possible to land a fixed wing drone on a net attached to the top of a car moving at $70 \, \mathrm{km \, h^{-1}}$ [19]. Note, however, that in this experiment the ground vehicle followed a linear trajectory, which is not always possible in SaR missions, where the debris forces the UGV to make turns almost continuously.

Indeed, there are substantial differences when trying to land a UAV on a terrestrial moving platform describing either a linear or a circular trajectory. Most of the research so far focused solely on the former, without thoroughly considering that the movement of the target can also be circular or even a mixture of both, thus producing random trajectories. This is, therefore, an interesting line of work, since in SaR tasks we want to provide the terrestrial robot with complete freedom of movement. In such a scenario, the UAV has to adapt to the trajectory described by the ground robot for a successful landing. The work presented in this paper takes a step forward in this direction by demonstrating a system capable of autonomously landing a UAV for both a linear and a circular trajectory of the moving landing platform.

Finally, all of the works above described strategies towards precise landing in moving platforms, but none presented a full system capable of operating for extended periods of time. In our work, a robust state machine together with a recovery and re-localization module allows for life-long operation, as we show in Section 4.

## 3. Proposed Approach

This section describes the proposed approach: (1) a state machine to execute robustly the complete autonomous takeoff, tracking, and landing of a UAV on a moving landing platform (Section 3.1); (2) detection and localization of the mobile target using a downward-looking camera (Section 3.2); and (3) vision-based tracking of the mobile platform while in flight (Section 3.3).

### 3.1. State Machine

The autonomous takeoff-tracking-landing system proposed in this paper builds upon a finite state machine (Figure 2) with five states: *landed*, *taking off*, *tracking*, *landing*, and *re-localizing*.

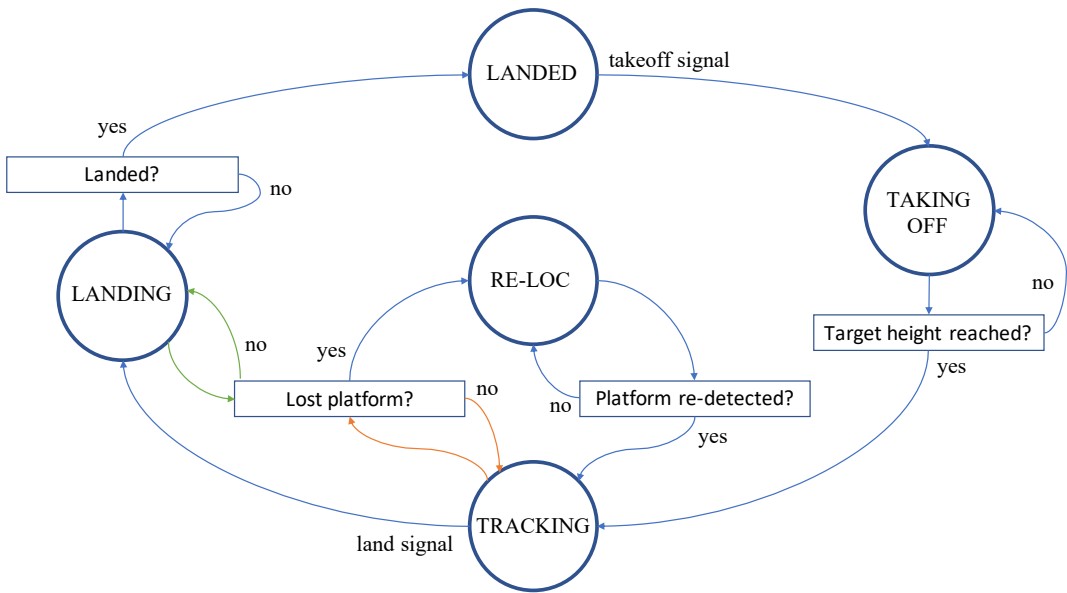

**Figure 2.** Finite state diagram that determines the behavior of the UAV.

*Landed* is the default state when launching the system and corresponds to the UAV resting on top of the landing platform, awaiting for a takeoff signal. As soon as this takeoff order is received, the drone's state changes to *taking off*, which represents the period during which the UAV is gaining altitude at a constant speed of $1\,\mathrm{m\,s^{-1}}$ along its z axis (please refer to Figure 1 for a view of the UAV's axis). The detection-localization algorithm (Section 3.2) is also launched at this point, as well as the tracking module (Section 3.3), so that the drone can start following the landing platform while ascending.

Once the nominal height has been reached (set to 4 m in our experiments), the state automatically changes to *tracking*, and the drone stops ascending. It will now follow the landing platform, keeping a constant altitude. To do so, a PID controller computes the necessary speed signals (both in the x and y axes) required to reduce the drone's distance to the landing platform's centroid in the xy-plane, which is parallel to the ground.

A land command (user-induced or automated) will trigger the start of the landing maneuver and shift the state to *landing*. The aerial robot will start its descent towards the moving landing platform at a constant downward speed of $-0.3\,\mathrm{m\,s^{-1}}$ along its z axis, while the height-adaptive PID controller provides the necessary speed commands along the UAV's x and y axis. Note that the PID gains are constantly being updated depending on the altitude (Section 3.3.1), thus height-adaptive.

**Recovery module.** The system can enter into recovery mode for either of the following two reasons: (1) the tracking algorithm registers when the landing platform was detected for the last time, and if more than 0.5 s pass without getting a new position, the state changes to *re-localizing*; (2) if the relative error between the landing platform's centroid and the UAV's body frame is bigger than a threshold (0.25 m in our experiments) at the final landing stages—when the sonar indicates values smaller than 0.7 m—the system will also enter recovery mode. In both cases, the drone will start gaining altitude at a speed of $1\,\mathrm{m\,s^{-1}}$, and the height-adaptive PID controller will be turned off so that no speed commands are sent along the x or y directions. Note that the second condition is designed to work as a more strict and early detection of potential landing failures.

The intuition behind ascending vertically is that the viewed area by the UAV's downward-looking camera is gradually increased. When the landing platform is viewed again, the state is changed to tracking and maintained so until the nominal tracking altitude of 4 m is reached again. At that point, new incoming landing signals may be processed again. As we will show in a set of extensive experiments (Section 4.2.3), this re-localization strategy will prove to be key when the landing platform moves faster than the nominal velocity, keeping the system alive and preventing failed landings.

### 3.2. Detection and Localization of the Mobile Platform

In this section, we describe the proposed method to *detect* and *localize* the landing platform relative to the UAV's coordinate frame. To this end, several standard computer vision techniques were used. It should be noted that this task is not the central topic of this work, but rather a required task to accomplish the rest of the steps that make up the autonomous landing of a UAV on a moving target. Thus, it was assumed that the landing platform had an easy to detect pattern (color-wise) or a marker on top of it. In either case, OpenCV [20] functions are readily available to accomplish shape- and color-based or more accurate marker-based detection.

The detection algorithm is summarized in Algorithm 1. First, the input frame is converted to the Hue, Saturation, Value (HSV) color model; second, a color mask is applied, where we keep all pixels within a certain HSV range, in particular that which corresponds to the red color; third, a Gaussian filter is applied to blur the image; finally, the Canny edge detection and Hough line transform algorithms are employed, followed by polygon fitting and computation of the centroid's coordinates in the image plane. An example result of this approach using the downward-looking camera of a real aerial vehicle (AR Drone 2.0) is shown in Figure 3.

---

**Algorithm 1** Detection algorithm.

Input: $video\_feed$

Output: $centroid$

1: **while** true **do**
2:　　$image \leftarrow getImage(video\_frame)$
3:　　$image\_hsv \leftarrow rgb2hsv(image)$
4:　　$masked\_image \leftarrow hsvMask(image\_hsv)$
5:　　$blurred\_image \leftarrow gaussianBlur(masked\_image)$
6:　　$edges \leftarrow cannyEdgeDetector(blurred\_image)$
7:　　$lines \leftarrow houghLineTransform(edges)$
8:　　$polygon \leftarrow polygonFitting(lines)$
9:　　$centroid \leftarrow computateFirstOrderMomentOfArea(polygon)$
10: **return** $centroid$

---

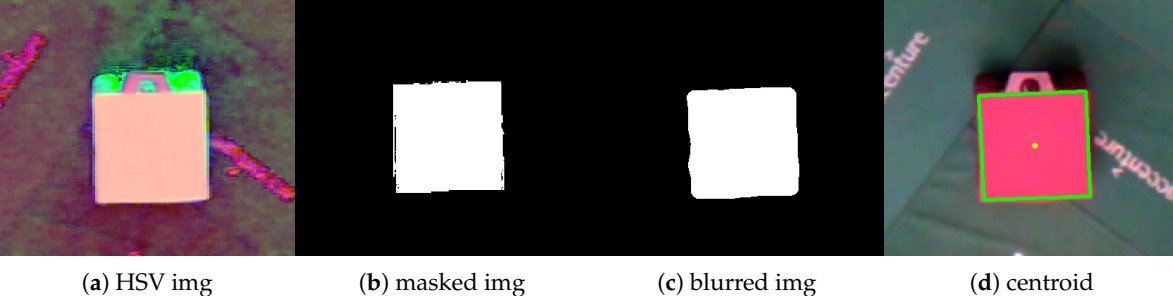

(**a**) HSV img　　　　(**b**) masked img　　　　(**c**) blurred img　　　　(**d**) centroid

**Figure 3.** Detection of the landing platform and extraction of its centroid using the imagery from the downward-looking camera of an AR Drone 2.0.

To convert the previously calculated coordinates of the centroid in the image plane $\Omega$ to a three-space vector in the camera frame the inverse of the well-known pinhole camera model can be used. This process is depicted in Figure 4.

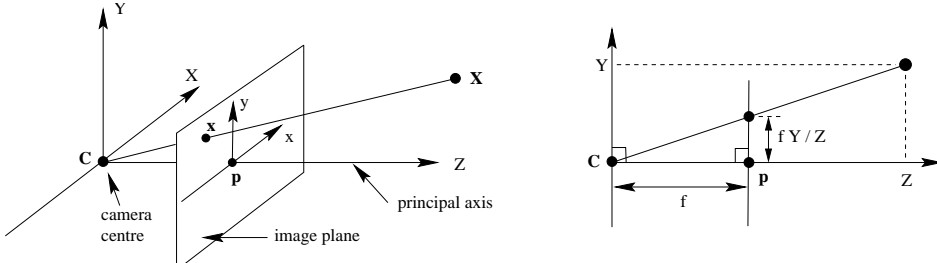

**Figure 4.** Pinhole camera geometry. *C* is the camera center and *p* the principal point [21].

Assuming radial distortion has been removed in a pre-processing step, Equations (1) and (2), i.e., the pinhole camera model equations, can be used to define the projection of a three-space vector in the camera frame into the image plane:

$$\lambda \begin{pmatrix} u' \\ v' \\ 1 \end{pmatrix} = \begin{pmatrix} f & 0 & c_x \\ 0 & f & c_y \\ 0 & 0 & 1 \end{pmatrix} \begin{pmatrix} X \\ Y \\ Z \end{pmatrix}, \tag{1}$$

$$\begin{aligned} u &= u'/\lambda \\ v &= v'/\lambda \end{aligned} \tag{2}$$

where $(u, v)^T$ is the projection of the point on the image plane expressed in pixels; $f$ is the focal length in pixels; $(c_x, c_y)^T$ are the coordinates of the principle point of the camera; and $(X, Y, Z)$ the three-space coordinates of the landing platform's centroid in the UAV's camera frame. Note that in the above equation we have assumed square and non-skewed pixels.

To obtain our desired 3D coordinates in the camera space, we need to invert the pinhole model. Additionally, if we know $Z$ beforehand (in our case, $Z$ corresponds to the vertical distance from the center of the UAV's camera frame to the center of the landing platform), we can easily compute $X$ and $Y$:

$$\begin{pmatrix} X \\ Y \end{pmatrix} = Z \begin{pmatrix} \frac{u - c_x}{f_x} \\ \frac{v - c_y}{f_y} \end{pmatrix} \tag{3}$$

and thus obtain the 3D position of the landing platform's centroid with respect to the UAV's camera frame.

We discard measurements taken when the Inertial Measurement Unit (IMU) indicates an inclination bigger than a threshold. Therefore, in theory, relative positions of the landing platform with respect to the UAV computed by the detection-localization algorithm are always obtained without a major tilt, thus producing reliable estimates to a certain degree. Nonetheless, a minor level of noise is always present in the output of this algorithm. Leveraging a Kalman filter, as we will explain in Section 3.3.2, will prove to be an effective way to deal with such measurement noise.

### 3.3. Tracking the Mobile Platform

Two variants for the tracking algorithm were explored. The first one uses the currently estimated 3D position of the landing platform's centroid relative to the UAV's body frame as the input cue for

a height-adaptive PID controller. The required 3D position is computed by transforming the output of the detection-localization algorithm presented in Section 3.2 to the UAV's body frame, as detailed in the following subsection. Note that by height-adaptive we refer to the fact that the PID gains are modified continuously depending on the UAV's flight altitude at every instant.

The second variant extends this height-adaptive PID with a Kalman filter to predict the future position of the landing platform. This prediction is then used as the target position for the same height-adaptive PID controller. The implementation of the Kalman filter is based on a previous work by the authors [22] where the prediction was used for tracking pedestrians.

Additionally, embedded in the tracking module, a *height control system* ensures that a proper descent speed is set in every instant depending on the current UAV's flight altitude. Furthermore, as mentioned above, this height-adaptive control system updates the PID gains depending on the current altitude: essentially, the goal is to have a faster response the closer the UAV is to the landing platform along the vertical axis (perpendicular to the ground).

After having presented the general scheme of both algorithms, i.e., with (w/) and without (w/o) the prediction of the future position of the moving platform, each variant is explained with more detail in Sections 3.3.1 and 3.3.2, respectively.

### 3.3.1. Height-Adaptive, Non-Predictive PID Controller

A PID controller is a control loop feedback mechanism to compute the necessary control variable $u(t)$ that makes the error $e(t)$ between the desired process value or setpoint $r(t)$ and the measured process value $y(t)$ converge to zero as fast as possible. To do so, it uses three different actions, namely a proportional, an integral, and a derivative action, each contributing differently to the control action. The mathematical expression of a PID controller is given by:

$$u(t) = K_p \, e(t) + K_i \int_0^t e(t') \, dt' + K_d \, \frac{\mathrm{d}e(t)}{\mathrm{d}t}, \tag{4}$$

where $e(t) = r(t) - y(t)$. In our system, the desired process value, $r(t)$, is the 2D zero vector, i.e., the origin of the UAV's coordinate frame relative to the coordinate frame itself. The measured process value, $y(t)$, is the two-vector containing the x and y coordinates from the 3D position of the landing platform's centroid relative to the UAV's body frame. Thus, $e(t)$ in our system is the distance in the xy-plane (parallel to the ground) between the UAV's frame and the centroid of the landing platform at time $t$.

Recall now that in the localization stage we compute the three-space coordinates of the landing platform in the drone's camera frame, which we denote by $P_{cam}$. If we are to use this position as a target value for the PID controller, the first step is to transform its coordinates to the UAV's body frame, $P_{body}$, since what we want is for the center of the UAV—and not for the center of its downward-looking camera—to get closer to the centroid of the landing platform. The transformation $T_{body\_cam}$ that maps a point from the camera frame to the body frame of the UAV is known from design and gives us:

$$P_{body} = T_{body\_cam} \, P_{cam}. \tag{5}$$

We can then compute the position error $e(t)$ using the expression:

$$e(t) = \sqrt{(P_{body_x})^2 + (P_{body_y})^2} \tag{6}$$

and finally, calculate the control variable $u(t)$.

**Tuning.** Finding the optimal PID gains (proportional gain $K_p$, integral gain $K_i$, and derivative gain $K_d$) was carried out through heuristic rules, i.e., looking first for the $K_p$ that provided the desired response while keeping $K_i$ and $K_d$ at zero; then gradually increasing $K_i$ to cancel the position error (and slightly

decreasing the $K_p$ so that the system did not become unstable); and, finally, increasing $K_d$ so that the response of the system was faster while fixing the values of $K_p$ and $K_i$ obtained in the former steps. In practice, we found a PI controller, i.e., without the derivative action, to work better than a complete PID.

**Height-adaptive PI controller.** We also found a height-adaptive PI controller to perform better than a fixed-gain PI, since different responses are needed when the drone is hovering at 4 m above the landing platform than moments before landing. In particular, the UAV needs to react faster the closer it is to the landing platform along the vertical axis.

An initial tuning of the PI gains for an altitude of 4 and 2 m in the simulation is shown in Table 1. Note that no distinction has been made between the x and y axes of the UAV, since we can disregard the different dynamics of the UAV with respect to each of these axes.

**Table 1.** Height-dependent PI gains.

| PID Gains | Altitude Range | |
|:---:|:---:|:---:|
| | **4 m** | **2 m** |
| $K_p$ | 0.694 | 0.697 |
| $K_i$ | 0.198 | 0.199 |

We noticed that an exponential function of the form:

$$K_x = Q\,e^{-T\,p_z} \tag{7}$$

could nicely fit the values we had obtained manually, where $Q$ and $T$ are the parameters of the function, $p_z$ the drone's altitude, and $K_x$ the height-dependent PID gain we want to model. After further fine-tuning, the final expressions for both the proportional and integral gains are the following:

$$K_p = 0.7\,e^{-0.002\,p_z},$$
$$K_i = 0.2\,e^{-0.002\,p_z}.$$

**Landing.** The descent speed during landing remains at a constant value of $0.3\,\mathrm{m\,s^{-1}}$ when flying 0.7 m above the landing platform, i.e., when the sonar indicates more than 0.7 m. Below 0.7 m, the UAV increases its downward speed notably to $2.0\,\mathrm{m\,s^{-1}}$. The intuition behind this design choice is that, when too close to the landing platform, the latter is not viewed completely by the UAV's downward-looking camera, and in turn the computed centroid might not represent the real center position of the platform. It is therefore a better option to rely on the correct measurements taken at altitudes higher than 0.7 m and then perform the final approach stages faster.

To determine whether the drone has successfully landed on the landing platform, we use the sonar measurements and the linear acceleration provided by the IMU. If there exists some linear acceleration, either in the x or y direction, that means the drone is moving. Moreover, if the sonar indicates a value smaller than a threshold for a certain period of time, the drone is assumed to have landed; this is not, however, a sufficient condition, since it might as well have landed on the ground. Therefore, to be certain that the UAV has actually landed on the moving platform and not on the ground, both conditions must be met, namely (1) the sonar measurement must be smaller than a threshold persistently and (2) the IMU must indicate a non-zero linear acceleration. Note that these assumptions are valid because the landing platform is assumed to be moving constantly.

In practice, the UAV mostly lands smoothly without major rebounds after the first contact with the landing platform, as we could verify in a set of preliminary experiments. After all, given that the height-adaptive PID controller ensures that the landing platform's centroid and the UAV's body frame are aligned along the vertical axis at all times, landing mostly occurs very close to the center of the

platform. Therefore, in all experiments (Section 4) we can safely use the initial touch point between the landing platform and the UAV as stop signal for data recording.

### 3.3.2. Height-Adaptive, Predictive PID Controller

In this section, we describe how we integrated the pedestrian trajectory prediction algorithm developed in a previous work by the authors [22] into the height-adaptive PID controller described in the previous section. The resulting pipeline looks as follows:

1.  The detection-localization algorithm (Section 3.2) outputs the centroid of the landing platform in the UAV's camera frame. The prediction algorithm requires as input a 3D point relative to an *inertial* reference system. Therefore, we must transform the centroid of the landing platform from the drone's camera frame into the the world's frame:

$$P_{world} = T_{world\_cam} \, P_{cam}. \tag{8}$$

2.  The new position is then sent to the prediction algorithm (Kalman filter), which returns a vector of future positions of the centroid of the landing platform relative to the world frame $\hat{P}_{world}$. The first element in this vector (with index zero) corresponds to the current position of the landing platform. The next element (index one) corresponds to the next predicted position after a user-defined time step. Correspondingly, the element with index two corresponds to a prediction carried out with twice the defined time step. In general, the number of steps in this path of predicted positions is computed as the ratio between a user-provided *path time* and the *time step*.
3.  Subsequently, the predicted future position of the landing platform is transformed from the world's frame into the UAV's body frame:

$$\hat{P}_{body} = T_{body\_world} \, \hat{P}_{world}. \tag{9}$$

4.  Finally, the x and y coordinates of $\hat{P}_{body}$ are used to calculate the controller's error, i.e., the distance in the xy-plane between the UAV and the predicted position of the landing platform. Using this error we can now calculate the speed commands in x and y, i.e., $u(t)$ in (4), that make this error converge to zero.

**Configuration of the Kalman filter.** We use a value of 0.1 s as the time step for our predictions and a path time of 0.1 s, thus obtaining a vector of future predictions of size two, where the element with index one corresponds to the predicted position of the landing platform. (Recall that the element with index zero corresponds to the *current* position of the landing platform's centroid.) We studied the effect of using different path times and then accessing different future positions within this vector of predictions depending on the altitude, but in a set of preliminary experiments, we found that the performance improvement of this design choice was minor or even negative.

By using the time step and path time defined above, if the landing platform were to move, for instance, linearly at a speed of $0.5 \, \text{m s}^{-1}$, the prediction algorithm would estimate an increment of 0.05 m along the current trajectory of the moving target. We found that such a minor prediction (in this example, the predicted position differs only in 5 cm from the current position) benefits the landing accuracy notably, as presented in Section 4, when compared to the non-predictive system.

For completeness, we detail the co-variance matrices of the process noise, $Q$, and the observation noise, $R$, both diagonal matrices of the form:

$$Q = \begin{pmatrix} 2 & 0 & 0 & 0 \\ 0 & 2 & 0 & 0 \\ 0 & 0 & 2 & 0 \\ 0 & 0 & 0 & 2 \end{pmatrix}, \qquad R = \begin{pmatrix} 2 & 0 \\ 0 & 2 \end{pmatrix}, \tag{10}$$

which were obtained empirically.

As will become apparent in Section 4, employing a Kalman filter to predict the future positions of the landing platform not only provided the system with knowledge about the trajectory described by the UGV, but also helped further stabilize the measurements from the detection-localization algorithm (Section 3.2).

In this section, we described a system for the autonomous takeoff, tracking, and landing of a small UAV on a moving landing platform. We presented two variants for the tracking module: a non-predictive height-adaptive PID controller and its predictive counterpart, which leverages a Kalman filter to predict the future position of the landing platform. By doing so, not only do we filter out noise in the measured relative 3D position of the landing platform with respect to the UAV, but also allow the UAV itself to stay slightly ahead of the UGV by directly feeding this virtual future position to the height-adaptive PID. This approach will prove to be crucial to accomplish successful landings, especially when the landing platform describes non-linear trajectories.

## 4. Results

In this section, we present the obtained results both in the simulated and real environments. In Section 4.1, some general design considerations are given. Section 4.2 details the results obtained for an extensive set of experiments in the simulated environment, which serve as a validation of the system performance both in nominal and more demanding conditions. Finally, in Section 4.3, we present some qualitative real-world experiments to demonstrate that our system can be deployed in real robotic platforms.

### 4.1. Design of the Testing Environments

As previously mentioned, the scope of this work is focused on the development of a robust control algorithm for the autonomous takeoff, tracking, and landing of a UAV on a moving landing platform for life-long operation. Therefore, the experiments were designed so as to facilitate the detection task. Nevertheless, it should be noted that the control algorithm can operate together with any kind of detection method.

The initial approach was based on placing a visual marker on top of the landing platform, as shown in Figure 1. However, this approach was discarded after testing that from a height of 4 m the downward-looking camera of the real AR Drone 2.0 was unable to detect the marker robustly. Therefore, the visual marker was replaced by a red-colored, square-like landing platform like that shown in Figures 3 and 5 for the real and simulated environments, respectively. Such a landing platform can be easily detected by using standard computer vision techniques based on color and shape detection, as already described in Section 3.2. To further simplify this task, both simulated and real tests were performed on flat and feature-less terrains.

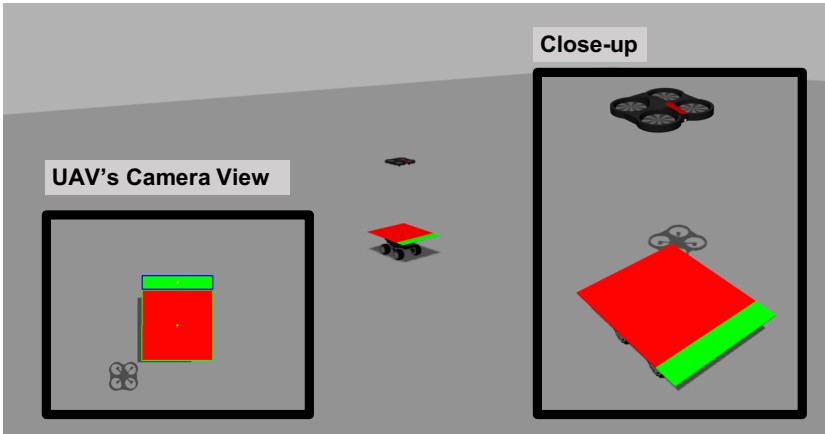

**Figure 5.** Simulated environment and robotic platforms.

*4.2. Experiments in the Simulated Environment*

The simulation experiments were conducted in the Gazebo simulator [8]. We designed two sets of experiments: one evaluated the system under the same initial conditions; a second set of tests aimed at evaluating the robustness of the system as a whole and the utility of our novel re-localization module in particular.

4.2.1. Experiments under the Same Initial Conditions

In this set of experiments, the system was re-launched every time so as to have the same initial conditions for every try. Moreover, the landing platform followed two different trajectories, namely linear and circular. A total of 20 takeoff-tracking-landing maneuvers were executed, 10 for each trajectory type, always re-initiating the whole system for every new test. Within a trajectory type, we tested the two variants of our tracking algorithm, i.e., w/ and w/o prediction. In all experiments, we computed the Euclidean distance between the landing platform's centroid and the UAV's body frame with respect to the latter. In the following, we will denote this as the *error*. Note that this error only coincides with the error fed to our height-adaptive PID controller in the mode w/o prediction. The details of the two types of trajectories together with the results obtained in each case are described next.

**Linear trajectory.** The UGV (and thus the landing platform) describes a linear trajectory at a speed of $0.5\,\mathrm{m\,s^{-1}}$ along its x axis. Initially, both the UGV and the UAV are at rest, with the latter lying on the landing platform. After a takeoff signal, the aerial vehicle begins its ascent phase to a pre-defined height of 4 m, and the UGV starts moving along a linear trajectory. Upon detection of the landing platform, the UAV automatically begins to follow the UGV by reducing its distance to the latter in the xy-plane, as explained in previous sections. After 30 s in tracking mode, a landing signal is sent automatically to the drone, which begins its descent towards the UGV.

Figure 6 shows the trajectory described by the UAV and the UGV for a single experiment of the linear trajectory. Results for both the non-predictive and predictive modes of the controller are presented. At first glance, no notable differences can be appreciated. The system w/ prediction is, however, more stable than its non-predictive counterpart as can be noted by visualizing the slightly more smooth curves in Figure 6b compared to those in Figure 6a. Importantly, note the scale of the y axis in Figure 6a,b and how the trajectory of the UAV is bounded within roughly 3 cm after the first 5 m in both cases (w/ and w/o prediction), following almost perfectly the linear trajectory described by the landing platform.

Figure 7 plots the error along both the x and y axis for a single experiment of the linear trajectory. As with the previous figures, results for both for the non-predictive and predictive variants are shown. Note that the error along the UAV's y axis is close to zero for both modes (w/ and w/o prediction), as was expected in the case of a linear trajectory. As for the error along the x axis, the general trend is similar in both modes, though more bounded for the algorithm w/ prediction. The big initial peak in Figure 7a is due to the fact that the UGV starts moving at the same time that the UAV takes off, thus generating a relatively large initial error along the x axis that is corrected after the first 10 s of simulation.

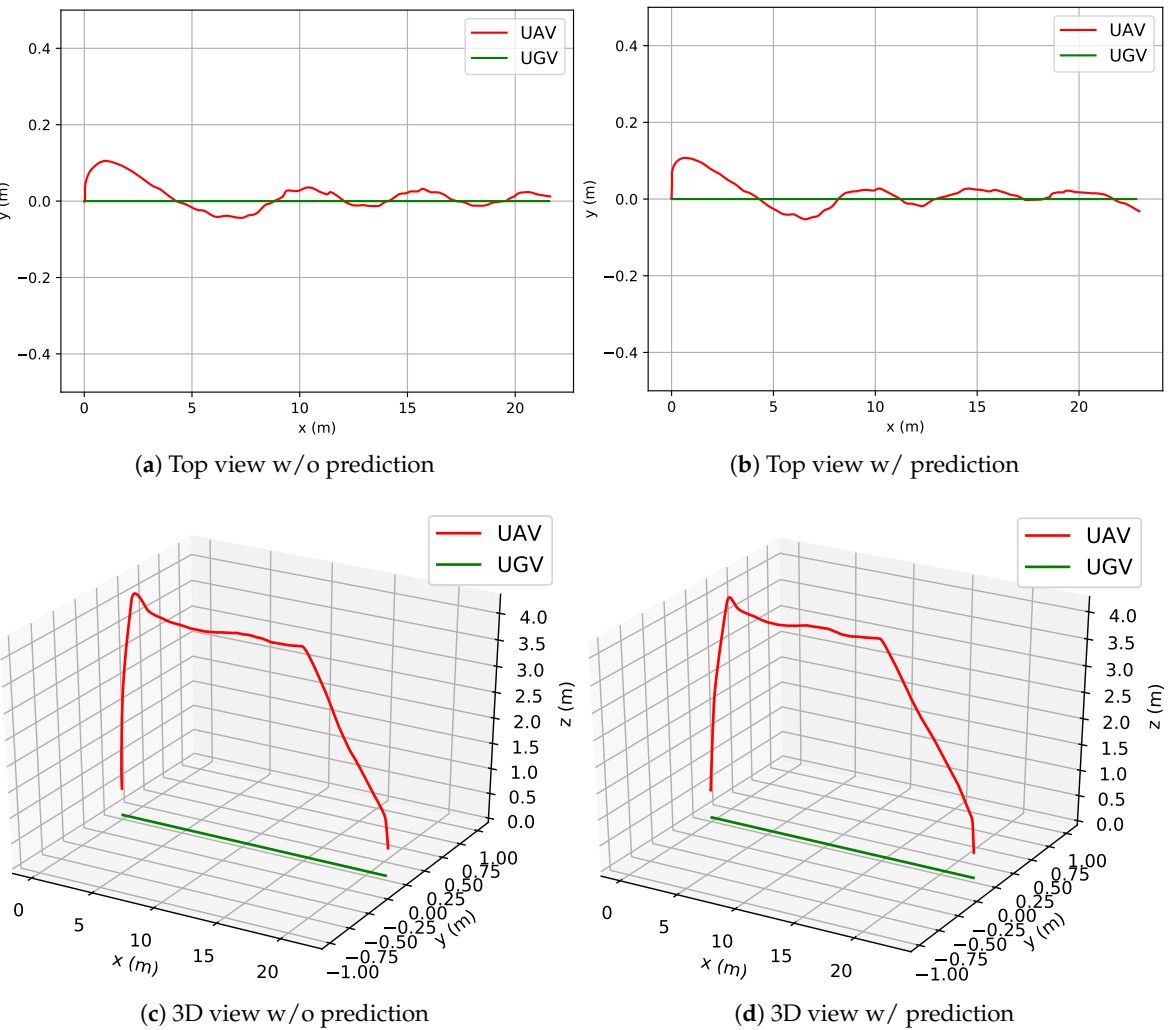

(**a**) Top view w/o prediction　　　　　　　　　　　　(**b**) Top view w/ prediction

(**c**) 3D view w/o prediction　　　　　　　　　　　　(**d**) 3D view w/ prediction

**Figure 6.** Views of the movements of the UGV and the UAV during a linear trajectory experiment.

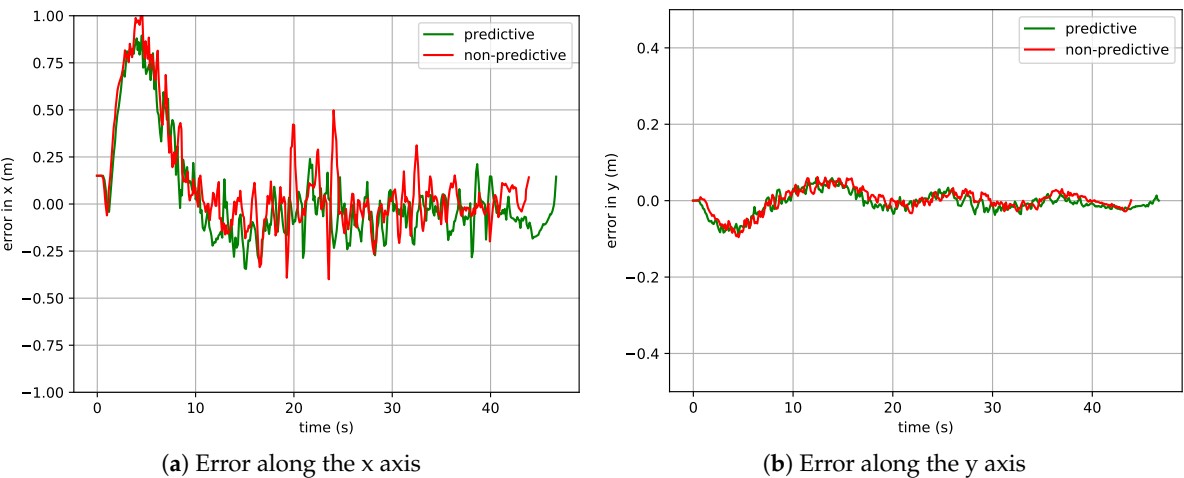

(**a**) Error along the x axis　　　　　　　　　　　　(**b**) Error along the y axis

**Figure 7.** Errors for a linear trajectory sequence.

**Circular trajectory.** In this experiment, the UGV describes circles with a forward speed along its x axis of $0.5\,\mathrm{m\,s^{-1}}$ and an angular velocity along its z axis of $0.05\,\mathrm{rad\,s^{-1}}$. In theory, this would result in a circular trajectory of radius 10 m. In practice, due to the friction coefficient of the UGV's wheels, it results in a circle of radius 12 m in this particular case. Note that the takeoff-tracking-landing procedure described for the linear trajectory is also followed here.

Figure 8 shows the trajectory described by the UAV and the UGV for a single experiment of the circular trajectory. As with previous figures, results for both the non-predictive and predictive variants are presented for comparison. Note how the UAV's trajectory resulting from the system w/ prediction matches that of the landing platform slightly better than in non-predictive mode, especially in the region between 6 and 8 m along the x axis.

Similarly, Figure 9 plots the error along both the x and y axis for a single experiment of the circular trajectory. Once again, non-predictive and predictive approaches are compared. The reader will again note that the system leveraging a predictive action outperforms its non-predictive counterpart in terms of a smaller error overall, both along the x and the y axis.

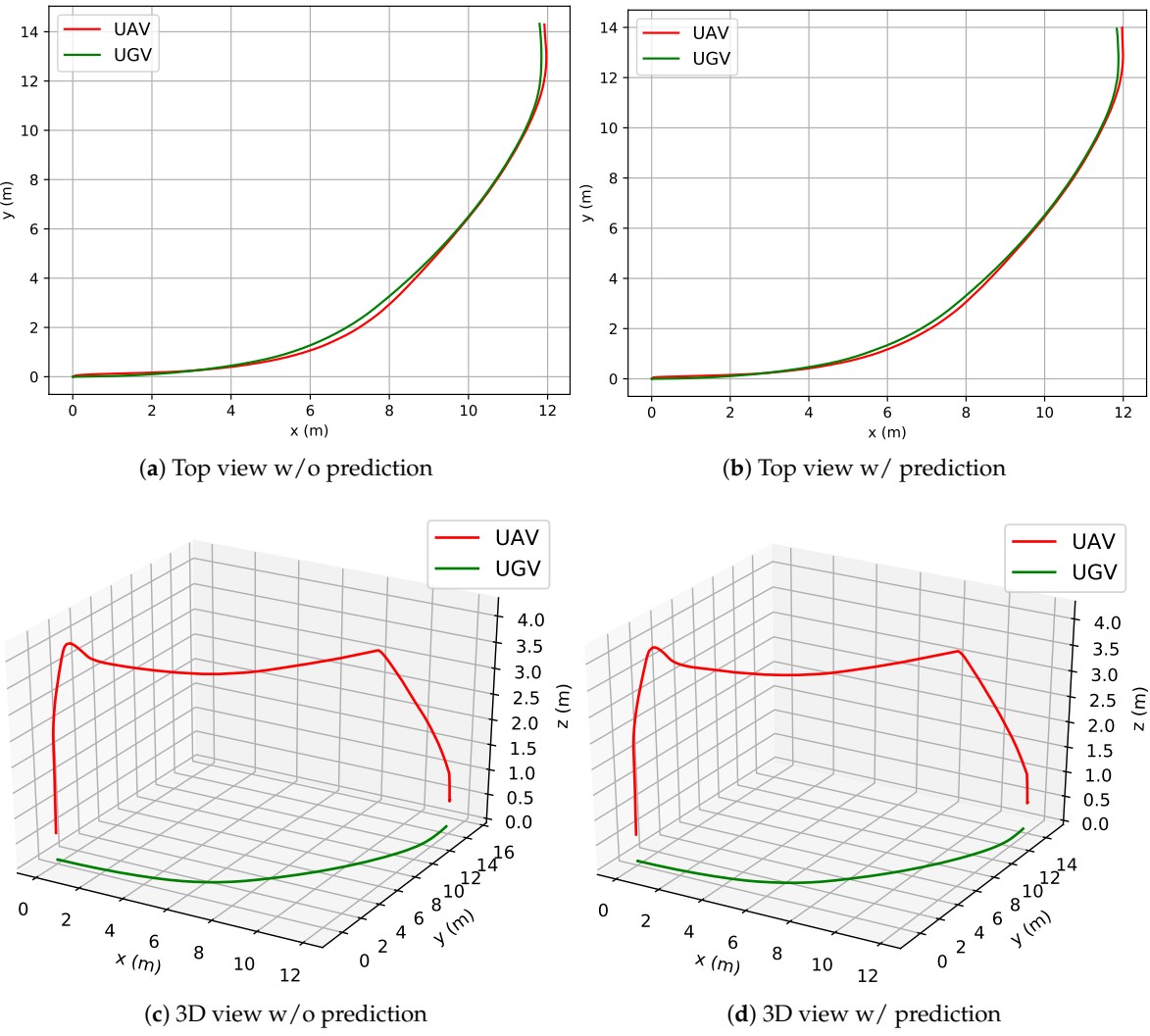

(**a**) Top view w/o prediction

(**b**) Top view w/ prediction

(**c**) 3D view w/o prediction

(**d**) 3D view w/ prediction

**Figure 8.** Views of the movements of the UGV and the UAV during a circular trajectory experiment.

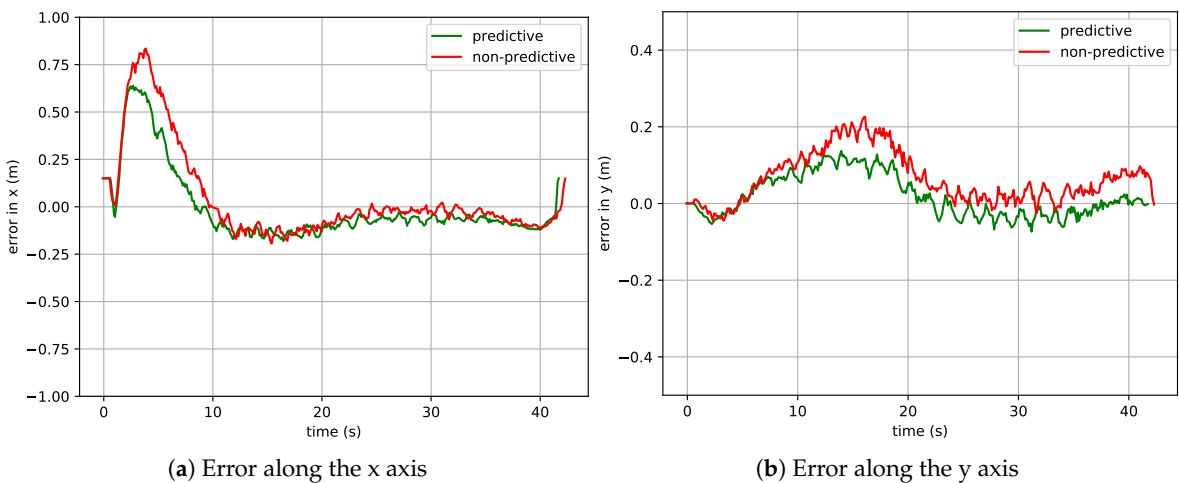

(**a**) Error along the x axis

(**b**) Error along the y axis

**Figure 9.** Errors for a circular trajectory sequence.

### 4.2.2. Overall Results in the Simulated Environment for a Linear and Circular Trajectory

Figure 10 reports the errors in the x and y axis, respectively, for all 20 experiments. Each box within a boxplot represents five experiments, where for each experiment we computed the mean error value of the whole trajectory (from takeoff until landing).

Note that the mean error in all cases was always lower than 0.12 m. Note as well how for both trajectory types (linear and circular) and for both the error in the x and y axes, the variant w/ prediction achieved lower error than its non-predictive counterpart.

In particular, the error along the x axis (Figure 10a) for a linear trajectory was around 0.10 cm on average for the system w/o prediction and around 0.04 cm for the system w/ predictive action; for a circular trajectory, this error was smaller in general terms: around 0.04 cm for the system w/o prediction and close to zero when employing the predictive height-adaptive PI controller.

With regards to the error along the y axis (Figure 10b), for a linear trajectory the mean error for all ten experiments (five for each mode, i.e., w/o and w/ prediction) was very small with almost no dispersion of the data at all. For the circular trajectory, the error produced by the non-predictive tracking algorithm was in the range of 0.06 cm, while the system leveraging the Kalman filter achieved lower errors in the range of 0.03 cm.

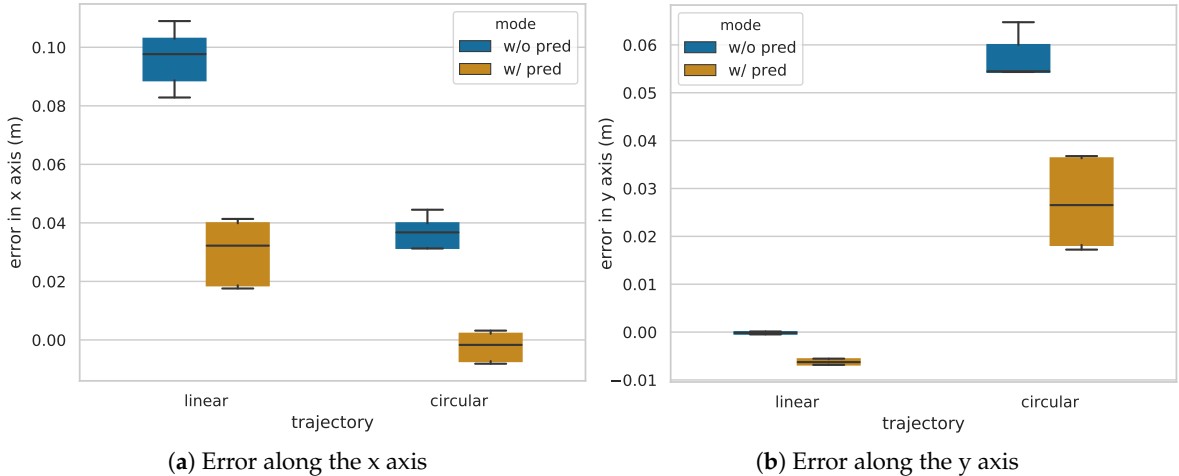

(**a**) Error along the x axis

(**b**) Error along the y axis

**Figure 10.** Comparison of the errors between linear and circular trajectories, both for the non-predictive (w/o pred) and predictive (w/ pred) variants.

Overall, the system performed robustly in all tests, managing to land flawlessly in all cases. We have therefore demonstrated how an approach based on a predictive height-adaptive PI controller outperforms its non-predictive counterpart, allowing the UAV to follow and land on the moving landing platform consistently, performing a more stable and accurate flight.

### 4.2.3. Experiments to Test the Life-Long Operation Capabilities of the System

To further demonstrate the robustness of our algorithm and its life-long operation capability, we carried out the following experiment: we launched the system once and let the UAV perform up to a maximum of 50 takeoff-tracking-landing maneuvers continuously. For every iteration, 10 s pass from the moment of takeoff before an automated landing signal is sent. Once the UAV lands, it rests for 1 s on top of the landing platform before taking off again to complete a new takeoff-tracking-landing maneuver.

This test was carried out for a circular trajectory and for two different velocity conditions of the landing platform, resulting in a total of 100 attempted maneuvers: on the one hand, nominal conditions ($v_x = 0.5 \, \mathrm{m \, s^{-1}}$, $w_z = 0.05 \, \mathrm{rad \, s^{-1}}$); and, on the other hand, more demanding conditions ($v_x = 0.7 \, \mathrm{m \, s^{-1}}$, $w_z = 0.07 \, \mathrm{rad \, s^{-1}}$). Note that, in practice, the UGV model, i.e., the Summit XL, rarely reaches linear speeds higher than $v_x = 0.7 \, \mathrm{m \, s^{-1}}$, commonly operating at a nominal speed of $0.5 \, \mathrm{m \, s^{-1}}$ along its x axis. However, we wanted to evaluate our re-localization module thoroughly under more challenging conditions. We used our best-performing system, namely our system based on a predictive, height-adaptive PID controller. Figure 11 visualizes the trajectories described by both the UAV and the UGV for the two speed conditions mentioned above.

Note that, in practice, the trajectory followed by the UGV is never exactly a circle due to the wheels' friction coefficients. Moreover, the UGV also drifted in time, thus resulting in various circles centered in different locations, as can be seen in Figure 11.

As gathered in Table 2, for a velocity of ($v_x = 0.5 \, \mathrm{m \, s^{-1}}$, $w_z = 0.05 \, \mathrm{rad \, s^{-1}}$) the system managed to land the aerial vehicle successfully in all 50 consecutive maneuvers with a Mean Absolute Error (MAE) along the UAV's x axis of 0.127 m and 0.245 m along its y axis. Moreover, the maximum absolute error along the x and y axes was 0.734 m and 0.653 m, respectively. When operating at a higher velocity ($v_x = 0.7 \, \mathrm{m \, s^{-1}}$, $w_z = 0.07 \, \mathrm{rad \, s^{-1}}$), the re-localization module had to be launched 16 times, leaving a total of 34 successful landings. In this case, the MAE along the UAV's x axis was 0.245 m and 0.232 m along the y axis, and the maximum absolute errors were 1.526 m and 1.441 m for the x and y axes, respectively.

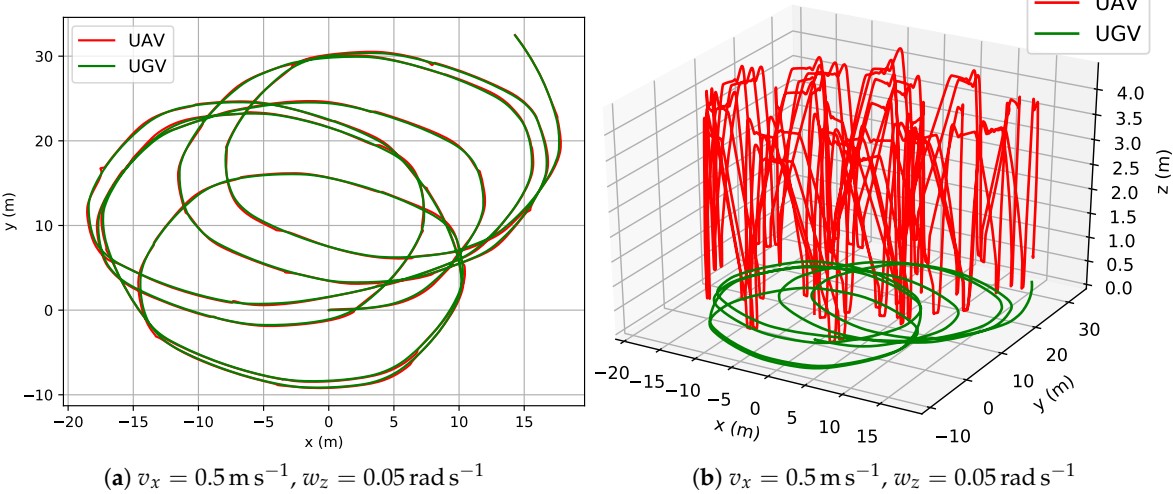

(**a**) $v_x = 0.5 \, \mathrm{m \, s^{-1}}$, $w_z = 0.05 \, \mathrm{rad \, s^{-1}}$          (**b**) $v_x = 0.5 \, \mathrm{m \, s^{-1}}$, $w_z = 0.05 \, \mathrm{rad \, s^{-1}}$

**Figure 11.** *Cont.*

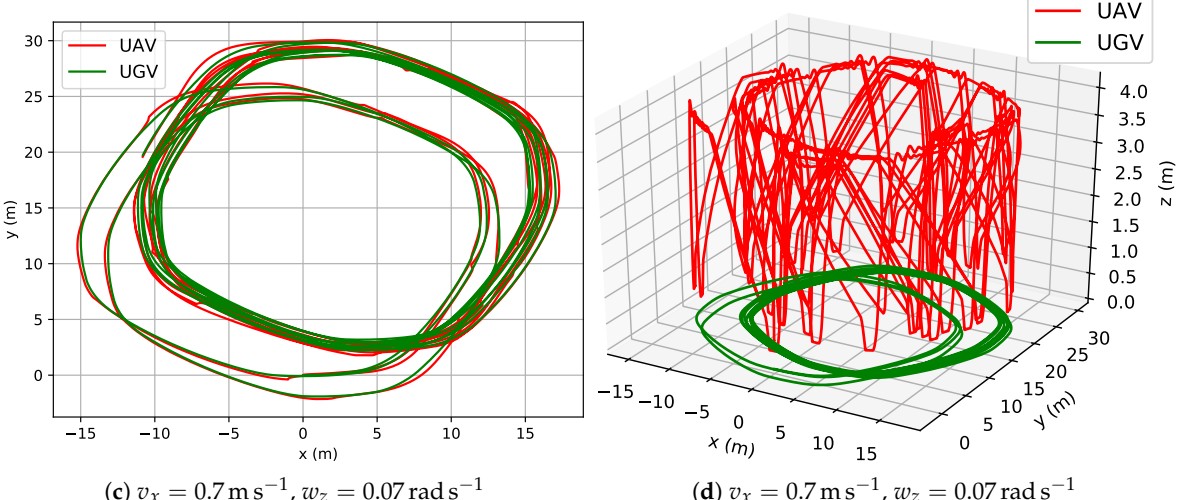

(c) $v_x = 0.7\,\mathrm{m\,s^{-1}}$, $w_z = 0.07\,\mathrm{rad\,s^{-1}}$      (d) $v_x = 0.7\,\mathrm{m\,s^{-1}}$, $w_z = 0.07\,\mathrm{rad\,s^{-1}}$

**Figure 11.** Trajectories described by the UAV (red) and the UGV (green) when performing a total of 50 consecutive takeoff-tracking-landing maneuvers continuously for two different linear and angular speeds of the UGV.

**Table 2.** Statistics for the life-long capability experiments. Linear velocities ($v_x$) are given in $\mathrm{m\,s^{-1}}$ and angular velocities ($w_z$) in $\mathrm{rad\,s^{-1}}$.

| Study Variables | Landing Platform Velocities $(v_x, w_z)$ | |
|---|---|---|
| | $(0.5, 0.05)$ | $(0.7, 0.07)$ |
| Total test time | 1072.41 s | 1249.09 s |
| Successful landings (num) | 50/50 | 34/50 |
| Successful landings (%) | 100% | 68% |
| Re-localization maneuvers | 0 | 16 |
| $\max(|error\_x|)$ | 0.734 m | 1.526 m |
| $MAE_x$ | 0.127 m | 0.245 m |
| $\max(|error\_y|)$ | 0.653 m | 1.441 m |
| $MAE_y$ | 0.103 m | 0.232 m |

Note that all 16 recovery maneuvers under the more challenging conditions were triggered not because the UAV lost sight of the landing platform, but due to a too large error—greater than 0.25 m in our experiments—in the moments before landing. Such an early detection of potential failed landings allows the system to avoid accidents and flawlessly continue functioning even under challenging velocity conditions of the landing platform. An example of a relocalization maneuver can be viewed in Figure 12. Moreover, Figure 13 shows the error along the UAV's z axis during the first 200 s of the experiment for both velocity conditions studied. Overall, the rate of successful landings was 100% for the nominal velocity of the UGV and 68% in the case of more challenging (and rather unusual) conditions.

In this subsection, we demonstrated how our system performs flawlessly at the nominal speed of the UGV. More importantly, we have shown that by leveraging our novel recovery module, potential failed landing maneuvers can be detected and avoided, thus demonstrating the great benefits that such a system brings about, both in terms of robustness and reliability.

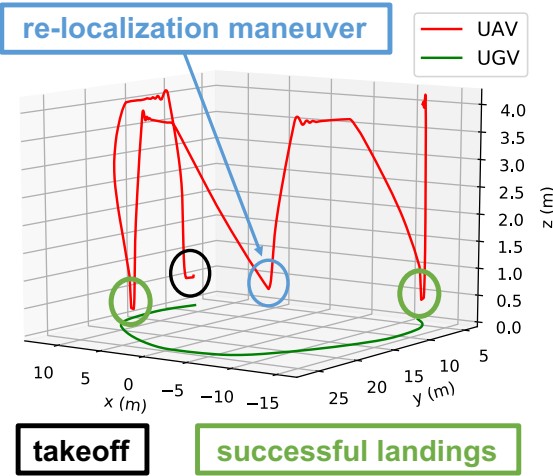

**Figure 12.** Detail of the 3D trajectory for a velocity of ($v_x = 0.7\,\text{m s}^{-1}$, $w_z = 0.07\,\text{rad s}^{-1}$) where a re-localization maneuver takes place. The run-time interval represented is [47.79 s, 99.84 s].

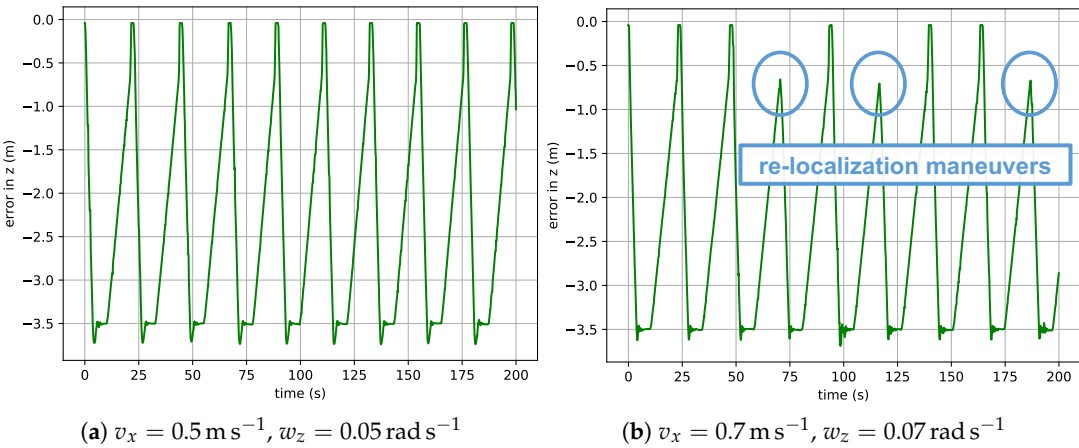

(**a**) $v_x = 0.5\,\text{m s}^{-1}$, $w_z = 0.05\,\text{rad s}^{-1}$    (**b**) $v_x = 0.7\,\text{m s}^{-1}$, $w_z = 0.07\,\text{rad s}^{-1}$

**Figure 13.** Error along the z axis (relative UAV-UGV distance along the z axis) during the first 200 s for both velocity conditions studied. (**b**) shows three re-localization maneuvers.

### 4.3. Experiments in the Real Environment

On the real robotic platforms we only tested the height-adaptive PID w/o predictive action, since for the predictive system to have worked we would have needed an additional means to localize the landing platform's position in global coordinates, as described in Section 3.3.2. In the simulated environment, transforming positions to a fixed global frame was straightforward. In the real world, however, this is more complex; implementing a Visual Inertial Odometry (VIO) or even a full visual Simultaneous Localization and Mapping (vSLAM) system would have been required in order to localize the drone in the scene with respect to a fixed frame.

What we did, however, was localize the landing platform relative to the UAV's coordinate frame, which is the only input required by the non-predictive approach. Therefore, on the real robotic platforms we qualitatively tested the system that employs our novel height-adaptive PID w/o prediction.

In particular, we performed five takeoff-tracking-landing sequences both for the linear and circular trajectory, following the same strategy as that described in Section 4.2.1. The UAV landed successfully in all five experiments for the linear trajectory and only failed once for the circular trajectory. The landing quality is visualized in Figure 14. We therefore demonstrate that the system presented in this work can be deployed on real robotic platforms. Figure 15 visualizes one of the linear trajectory experiments, and Figure 16 shows a sequence of a recovery maneuver. The complete sequences can be found in the video provided as Supplementary Material, or at https://youtu.be/CCrPBw_we2E.

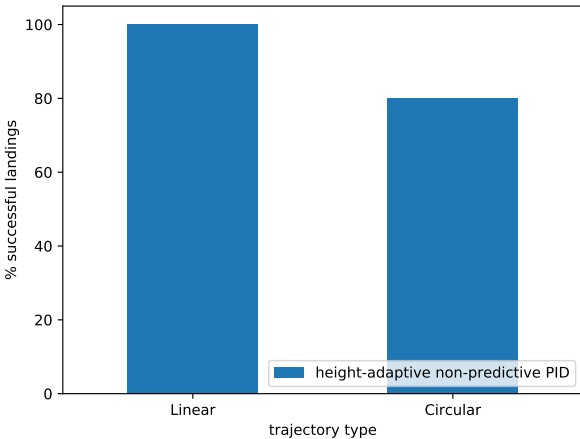

**Figure 14.** Percentage of successful landings in the real environment when using the height-adaptive, non-predictive PID controller for a linear and circular trajectory of the UGV. Note that these experiments were obtained by re-launching the system from scratch for every new test, as depicted in Section 4.2.1.

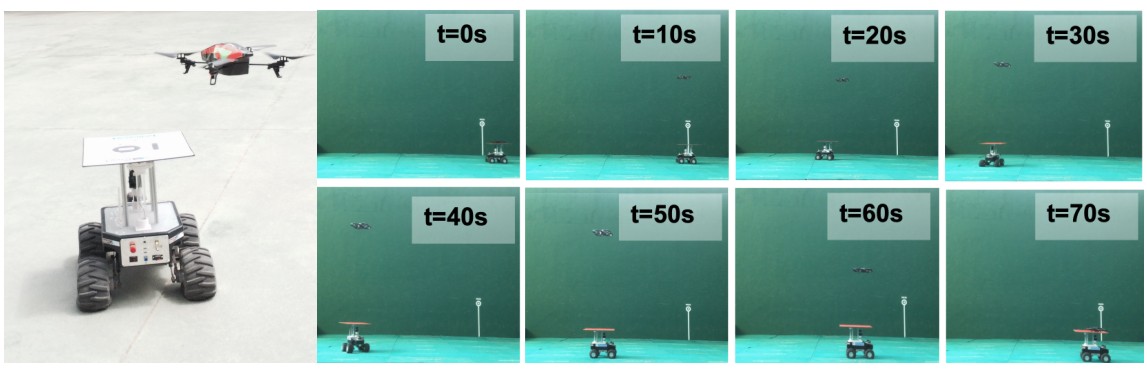

(**a**) Real robots (**b**) Landing sequence of the real UAV

**Figure 15.** Real robotic platforms (**a**) and landing sequence (**b**).

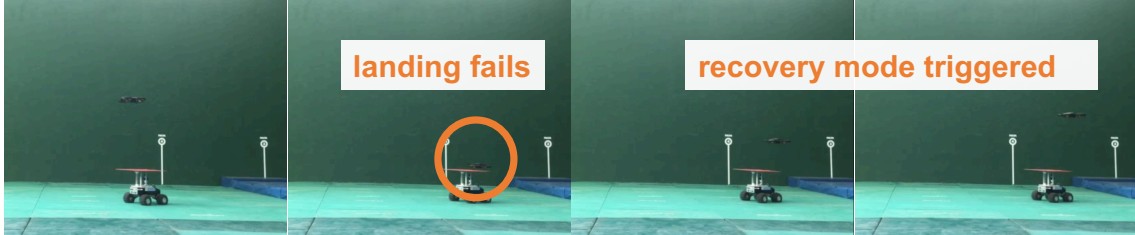

**Figure 16.** Re-localization maneuver in the real environment.

The reader must note that the real experiments were targeted as a qualitative demonstration of how our system can be integrated into real robotic platforms. We believe that the numerous quantitative experiments presented for the simulated environment (where we have used the same UAV model as in the real tests, as well as the same UGV) can serve to demonstrate the robustness and accuracy of the system, while the qualitative tests performed on the real robots can demonstrate that our system can be deployed on the real world.

## 5. Conclusions

In this work, we proposed a ROS-based system that enables a UAV to take off, track, and land autonomously on a moving landing platform. A novel height-adaptive PID controller suffices to operate the UAV satisfactorily when the landing platform describes either a linear or a circular trajectory at a speed of $0.5\,\mathrm{m\,s^{-1}}$ along its x axis. Introducing a Kalman filter to predict the future

position of the landing platform further improves the overall performance of the system, reducing the position error in comparison to the non-predictive approach.

Furthermore, we proposed a finite state machine architecture to keep track of different stages robustly. Together with a novel recovery module, they enable our system to operate in a continuous manner, providing it with life-long operation capability.

We extensively tested the system in the simulated environment (Gazebo), executing a total of 120 takeoff-tracking-landing sequences and reporting detailed results that validate the system's performance. We also implemented our algorithms on real robotic platforms and carried out qualitative evaluations, thus demonstrating that our system can be deployed in the real world.

Regarding future work, using a UAV with a better downward-looking camera would allow leveraging a marker detection system instead of the current color- and shape-based detection algorithm. By doing so, the whole system could be deployed in any kind of environment, regardless of the terrain's texture. Furthermore, a module could be added to localize the UAV in global coordinates, e.g., VIO or visual SLAM. This would allow implementing the predictive variant of our system in real platforms, which has demonstrated to outperform its non-predictive counterpart in the simulated environment.

**Supplementary Materials:** The software presented in this work is publicly available at https://github.com/pablorpalafox/uav-autonomous-landing. A video demonstrating the system can be viewed at https://youtu.be/CCrPBw_we2E. Furthermore, we also provide as Supplementary Material all our log files as raw CSV files (plus several Python scripts) so that the results presented in this work can be reproduced.

**Author Contributions:** Conceptualization, P.R.P. and M.G.; methodology, P.R.P., M.G., and J.V.; software, P.R.P. and M.G.; validation, P.R.P. and M.G.; formal analysis, P.R.P., J.V., and J.J.R.; investigation, P.R.P.; resources, P.R.P. and A.B.; data curation, P.R.P.; writing, original draft preparation, P.R.P.; writing, review and editing, M.G., J.V., and J.J.R.; visualization, P.R.P. and J.V., ; supervision, M.G.; project administration, A.B.; funding acquisition, J.J.R. and A.B.

**Funding:** The research leading to these results received funding from RoboCity2030-DIH-CM, Madrid Robotics Digital Innovation Hub, S2018/NMT-4331, funded by "Programas de Actividades I+Den la Comunidad de Madrid" and cofunded by Structural Funds of the EU, and from the project DPI2014-56985-R (Robotic protection of critical infrastructures), financed by the Ministry of Economy and Competitiveness of the Government of Spain.

**Conflicts of Interest:** The authors declare no conflict of interest.

## Abbreviations

The following abbreviations are used in this manuscript:

| | |
|---|---|
| UAV | Unmanned Aerial Vehicle |
| UGV | Unmanned Ground Vehicle |
| PID | Proportional-Integral-Derivative |
| 3D | Three-Dimensional |
| IBVS | Image-Based Visual Servoing |
| ROS | Robot Operating System |
| MPC | Model Predictive Control |
| VTOL | Vertical Take-Off and Landing |
| HSV | Hue, Saturation, Value |
| IMU | Inertial Measurement Unit |
| MAE | Mean Absolute Error |
| CSV | Comma-Separated Values |

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
