# Peer review of "Robust Visual-Aided Autonomous Takeoff, Tracking, and Landing of a Small UAV on a Moving Landing Platform for Life-Long Operation"

_applsci, doi:10.3390/app9132661_

Round 1

Reviewer 1 Report

line 297, simple should be simplify. 

2. line 280, please revisit sentence. 

3. Are you recording the initial touch point or the final point after the drone rests? As the drone could bounce on the platform and that can make quite some difference.

4. Is the target/drone relative position compensated with drone attitude? As a moving drone with tilted attitude could result in positioning error without compensation.

5. In real, what device measures the position error?

6. As predicted trajectory is one of the major contribution, can you put a little more words on it?

7. Great job!

Author Response

Response to Reviewer 1 Comments

Dear reviewer, first of all we would like to thank you for the valuable comments received. We are fully convinced that these comments made it possible to substantially improve the previous version of the manuscript.

Point 1: line 297, simple should be simplify. 

Response 1: Typo corrected. 

Point 2: line 280, please revisit sentence. 

Response 2: Typo corrected.

Point 3: Are you recording the initial touch point or the final point after the drone rests? As the drone could bounce on the platform and that can make quite some difference. 

Response 3: The bouncing effect was actually studied during a series of previous experiments. We noticed how, in practice, there are no major rebounds after the first contact with the landing platform. Nevertheless, since at all times the height-adaptive PID controller ensures that the landing platform's centroid and the UAV's body frame are aligned along the vertical axis, the landing occurs almost at the center of the platform every time. We can therefore safely record the initial touch point. We have tried to explain this effect and the conditions for a successful landing towards the end of section 3.3.1. (within the subsection "Landing") so that future readers can understand this point more clearly.

Point 4: Is the target/drone relative position compensated with drone attitude? As a moving drone with tilted attitude could result in positioning error without compensation. 

Response 4: We do discard measurements taken when the IMU indicates an inclination bigger than a threshold. Therefore, in theory, relative positions of the landing platform with respect to the UAV (computed by the detection-localization algorithm) are always obtained without a major tilt, thus producing reliable measurements.

Point 5: In real, what device measures the position error? 

Response 5: In the real robots we only tested the height-adaptive PID without predictive action, since for the system with prediction capabilities we would have needed an additional means to localize the landing platform's position in global coordinates. What we can do, however, is localize the landing platform relative to the UAV's coordinate frame. This we do using our detection-localization algorithm. Essentially, if we know the vertical distance between the drone and the landing platform, we can un-project the pixel coordinates of the landing platform's centroid back to 3D. By doing so, we obtain the 3D position of the landing platform with respect to the UAV's camera frame. We can then reference this position with respect to the UAV's body frame (since the transformation from camera to body in the UAV is known). Finally, we extract the X and Y coordinates of the position of the moving platform with respect to the UAV's body frame and use these for our height-adaptive PID.

So, in summary, for the height-adaptive PID with prediction we do need to express the landing platform's position in fixed global coordinates, while the system without prediction can work with the relative position of the landing platform with respect to the UAV. Since in the real environment we do not have a means to express robustly the landing platform's position in fixed global coordinates, we only test the non-predictive approach.

We have tried to also explain this further in Section 4.3.

Point 6: As predicted trajectory is one of the major contributions, can you put a little more words on it? 

Response 6: We have explained this in more depth in Section 3.2.2. We have detailed how the prediction algorithm that we make use of (developed in a previous work by Garzon) is designed and what parameters need to be tweaked. In particular, we have described how this implementation of a Kalman filter generates a path of future positions (each with a different time step). We also explain that we just extract the first predicted position of this vector, after having studied that using predictions with different time steps depending on the UAV's altitude did not bring about improvements in the overall performance.

Point 7: Great job!

Response 7: Thank you! We appreciate it.

Thank you again for your time and your valuable comments. We hope we have addressed all comments successfully.

Best regards,

The Authors

Reviewer 2 Report

It is work regarding the precision landing of multi-copter UAV on the moving platform. There are numerous ambiguous parts through the paper as follows.

On p.13, Section 4.3, please present plots of real experiment results which authors performed five times for linear and five times for circular trajectory. Also, please compare data from simulation and experiment and show us the main difference between the two cases.

Overall, authors used the combined method of PID controller for the altitude and KF algorithm for the position, but this is a generally used method and all related methods and source codes are already opened to the public from a long time ago. Also, although authors persist that the re-localization module is their third contribution in this manuscript, the re-localization module is not clearly described through the paper. In this sense, this reviewer thinks that this manuscript has low contributions in the precision landing area.

Author Response

Response to Reviewer 2 Comments

Dear reviewer, we would like to thank you for the comments received, as they were of great relevance in the revision process of our manuscript. Please find each of them addressed below.

Point 1: On p.13, Section 4.3, please present plots of real experiment results which authors performed five times for linear and five times for circular trajectory. Also, please compare data from simulation and experiment and show us the main difference between the two cases.

Response 1: While the numerous experiments carried out in the simulated environment were targeted as a means to gather a large amount of experimental data, both to adjust and validate the system's correct performance, we designed a few qualitative real experiments mainly to demonstrate that our system can also function on the real robotic platforms. We believe that the quantitative experiments presented for the simulated environments (where we have used the same UAV model as in the real tests, as well as the same UGV) can therefore demonstrate the robustness and accuracy of the system as a whole. We have also added this explanation at the end of Section 4.3.

Additionally, we present a new plot visualizing the landing quality as a percentage of successful landings in the real system for both linear and circular trajectories.

Finally, we also present a sequence of images depicting a recovery maneuver in the real system. The complete sequence visualizing this re-localization maneuver can be found in the new updated video that we present as additional material.

Point 2: Overall, authors used the combined method of PID controller for the altitude and KF algorithm for the position, but this is a generally used method and all related methods and source codes are already opened to the public from a long time ago.

Response 2: We introduce a novel height-adaptive PID controller, meaning that it adapts its parametrization and behavior depending on the UAV’s flight altitude. This height-adaptive PID is not used for altitude, but for position control in the xy-plane. It is further enhanced with a prediction module. Both approaches, however, are used for position control in the xy-plane in order to align both the UAV and the landing platform in the same vertical axis at all times.

Regarding already publicly available code, to the best of our knowledge this is the first work that proposes an integral system for life-long operation, offering a clean and parametrized framework integrated in ROS, which future researchers can easily tweak and adapt to their own works. As an example of to what extent our system presents a readily available framework for UAV-UGV cooperation, the user can even shift from the system with prediction to the system without prediction by a simple click on the user GUI, which pops up when launching the system. Additionally, the system is designed so that the user can always signal the UAV when to take off or when to land, again aiming to present a system readily available for Search and Rescue missions.

All in all, we believe that such a complete system, integrating a state machine for robust and life-long operation, a recovery and re-localization mode to prevent failed landings and a predictive control into a unified ROS-based system is not available publicly.

Point 3: Also, although authors persist that the re-localization module is their third contribution in this manuscript, the re-localization module is not clearly described through the paper. In this sense, this reviewer thinks that this manuscript has low contributions in the precision landing area.

Response 3: We have tried to explain the re-localization module in more depth (Section 3.1). Moreover, we have also emphasized a fourth contribution that was not properly highlighted in the first version of the manuscript, namely, our robust state machine, which allows the system to operate continuously for long periods of time. We hope this can help the community towards life-long operation in Search and Rescue tasks.

To further demonstrate the robustness of our algorithm and how it has been designed for life-long usage, we have included new experiments: we launch the system once and let the UAV perform up to a maximum of 50 takeoff-tracking-landing maneuvers continuously. We carry out these experiments for two different velocities of the UGV:

-       Under nominal conditions, the Summit XL (UGV) operates at a velocity of (vx = 0.5 m/s, wz = 0.05 rad/s). When carrying out the above described experiments under such nominal conditions the system never resorts to the recovery module and lands successfully in all 50 maneuvers.

-       To evaluate and demonstrate the benefits of introducing a recovery mode, we test the system at a higher velocity of (vx = 0.7 m/s, wz = 0.7 rad/s).  Under such more demanding conditions the re-localization strategy helps detect in advance possible failed landing maneuvers by evaluating the relative UAV-UGV error in the moments before landing: if the error surpasses a threshold, the recovery mode is activated, and the landing maneuver is postponed.

Furthermore, as mentioned in a previous answer, we also present a sequence of images depicting a recovery maneuver in the real system. The complete sequence of this re-localization maneuver can be found in the new updated video that we present as additional material.

Thank you again for your time and your valuable comments. We hope we have addressed all comments successfully.

Best regards,

The Authors

Round 2

Reviewer 2 Report

It is work regarding the precision landing of multi-copter UAV on the moving platform. There are numerous ambiguous parts through the paper as follows.

On p.13, Section 4.3, please present plots of real experiment results which authors performed five times for linear and five times for circular trajectory. Also, please compare data from simulation and experiment and show us the main difference between the two cases.

Ø  Response 1: While the numerous experiments carried out in the simulated environment were targeted as a means to gather a large amount of experimental data, both to adjust and validate the system's correct performance, we designed a few qualitative real experiments mainly to demonstrate that our system can also function on the real robotic platforms. We believe that the quantitative experiments presented for the simulated environments (where we have used the same UAV model as in the real tests, as well as the same UGV) can therefore demonstrate the robustness and accuracy of the system as a whole. We have also added this explanation at the end of Section 4.3.

Additionally, we present a new plot visualizing the landing quality as a percentage of successful landings in the real system for both linear and circular trajectories.

Finally, we also present a sequence of images depicting a recovery maneuver in the real system. The complete sequence visualizing this re-localization maneuver can be found in the new updated video that we present as additional material.

>> Satisfied

Overall, authors used the combined method of PID controller for the altitude and KF algorithm for the position, but this is a generally used method and all related methods and source codes are already opened to the public from a long time ago.

Ø  Response 2: We introduce a novel height-adaptive PID controller, meaning that it adapts its parametrization and behavior depending on the UAV’s flight altitude. This height-adaptive PID is not used for altitude, but for position control in the xy-plane. It is further enhanced with a prediction module. Both approaches, however, are used for position control in the xy-plane in order to align both the UAV and the landing platform in the same vertical axis at all times.

Regarding already publicly available code, to the best of our knowledge this is the first work that proposes an integral system for life-long operation, offering a clean and parametrized framework integrated in ROS, which future researchers can easily tweak and adapt to their own works. As an example of to what extent our system presents a readily available framework for UAV-UGV cooperation, the user can even shift from the system with prediction to the system without prediction by a simple click on the user GUI, which pops up when launching the system. Additionally, the system is designed so that the user can always signal the UAV when to take off or when to land, again aiming to present a system readily available for Search and Rescue missions.

All in all, we believe that such a complete system, integrating a state machine for robust and life-long operation, a recovery and re-localization mode to prevent failed landings and a predictive control into a unified ROS-based system is not available publicly.

>> Satisfied

Also, although authors persist that the re-localization module is their third contribution in this manuscript, the re-localization module is not clearly described through the paper. In this sense, this reviewer thinks that this manuscript has low contributions in the precision landing area.

Ø  Response 3: We have tried to explain the re-localization module in more depth (Section 3.1). Moreover, we have also emphasized a fourth contribution that was not properly highlighted in the first version of the manuscript, namely, our robust state machine, which allows the system to operate continuously for long periods of time. We hope this can help the community towards life-long operation in Search and Rescue tasks.

To further demonstrate the robustness of our algorithm and how it has been designed for life-long usage, we have included new experiments: we launch the system once and let the UAV perform up to a maximum of 50 takeoff-tracking-landing maneuvers continuously. We carry out these experiments for two different velocities of the UGV:

- Under nominal conditions, the Summit XL (UGV) operates at a velocity of (vx = 0.5 m/s, wz = 0.05 rad/s). When carrying out the above described experiments under such nominal conditions the system never resorts to the recovery module and lands successfully in all 50 maneuvers.

- To evaluate and demonstrate the benefits of introducing a recovery mode, we test the system at a higher velocity of (vx = 0.7 m/s, wz = 0.7 rad/s). Under such more demanding conditions the re-localization strategy helps detect in advance possible failed landing maneuvers by evaluating the relative UAV-UGV error in the moments before landing: if the error surpasses a threshold, the recovery mode is activated, and the landing maneuver is postponed.

Furthermore, as mentioned in a previous answer, we also present a sequence of images depicting a recovery maneuver in the real system. The complete sequence of this re-localization maneuver can be found in the new updated video that we present as additional material.

>> Satisfied